# *IGSF10* mutations dysregulate gonadotropin-releasing hormone neuronal migration resulting in delayed puberty

Sasha R Howard[†,1], Leonardo Guasti[†,1], Gerard Ruiz-Babot[1], Alessandra Mancini[1], Alessia David[2], Helen L Storr[1], Lousie A Metherell[1], Michael JE Sternberg[2], Claudia P Cabrera[3,4], Helen R Warren[4,5], Michael R Barnes[3,4], Richard Quinton[6], Nicolas de Roux[7,8,9], Jacques Young[10,11,12,13], Anne Guiochon-Mantel[10,11,12], Karoliina Wehkalampi[14], Valentina André[15], Yoav Gothilf[16], Anna Cariboni[15,17] & Leo Dunkel[1,*]

## Abstract

Early or late pubertal onset affects up to 5% of adolescents and is associated with adverse health and psychosocial outcomes. Self-limited delayed puberty (DP) segregates predominantly in an autosomal dominant pattern, but the underlying genetic background is unknown. Using exome and candidate gene sequencing, we have identified rare mutations in *IGSF10* in 6 unrelated families, which resulted in intracellular retention with failure in the secretion of mutant proteins. *IGSF10* mRNA was strongly expressed in embryonic nasal mesenchyme, during gonadotropin-releasing hormone (GnRH) neuronal migration to the hypothalamus. *IGSF10* knockdown caused a reduced migration of immature GnRH neurons *in vitro*, and perturbed migration and extension of GnRH neurons in a gnrh3:EGFP zebrafish model. Additionally, loss-of-function mutations in *IGSF10* were identified in hypothalamic amenorrhea patients. Our evidence strongly suggests that mutations in *IGSF10* cause DP in humans, and points to a common genetic basis for conditions of functional hypogonadotropic hypogonadism (HH). While dysregulation of GnRH neuronal migration is known to cause permanent HH, this is the first time that this has been demonstrated as a causal mechanism in DP.[‡]

**Keywords** delayed puberty; GnRH; hypothalamic amenorrhea; neuronal migration; puberty
**Subject Categories** Development & Differentiation; Genetics, Gene Therapy & Genetic Disease; Urogenital System

## Introduction

Puberty is the critical developmental stage during which reproductive capacity is attained. The onset of puberty is driven by the reactivation of the hypothalamic–pituitary–gonadal (HPG) axis after relative quiescence during childhood, with an increase in the pulsatile release of gonadotropin-releasing hormone (GnRH). While the timing of pubertal onset varies within and between different

1   Centre for Endocrinology, William Harvey Research Institute, Barts and the London School of Medicine and Dentistry, Queen Mary University of London, London, UK
2   Centre for Integrative Systems Biology and Bioinformatics, Department of Life Sciences, Imperial College London, London, UK
3   Centre for Translational Bioinformatics, William Harvey Research Institute, Barts and the London School of Medicine and Dentistry, Queen Mary University of London, London, UK
4   NIHR Barts Cardiovascular Biomedical Research Unit, Queen Mary University of London, London, UK
5   Department of Clinical Pharmacology, William Harvey Research Institute, Barts and The London School of Medicine, Queen Mary University of London, London, UK
6   Institute of Genetic Medicine University of Newcastle-upon-Tyne, Newcastle-upon-Tyne, UK
7   Unité Mixte de Recherche 1141, Institut National de la Santé et de la Recherche Médicale, Paris, France
8   Université Paris Diderot, Sorbonne Paris Cité, Hôpital Robert Debré, Paris, France
9   Laboratoire de Biochimie, Assistance Publique-Hôpitaux de Paris, Hôpital Robert Debré, Paris, France
10  Univ Paris-Sud, Le Kremlin Bicêtre, France
11  INSERM UMR-1185, Le Kremlin Bicêtre, France
12  Assistance Publique-Hôpitaux de Paris, Bicêtre Hospital, Le Kremlin-Bicêtre, France
13  Department of Reproductive Endocrinology, Bicêtre Hospital, Le Kremlin-Bicêtre, France
14  Children's Hospital, Helsinki University Hospital and University of Helsinki, Helsinki, Finland
15  Department of Pharmacological and Biomolecular Sciences, University of Milan, Milan, Italy
16  Department of Neurobiology, The George S. Wise Faculty of Life Sciences and Sagol School of Neuroscience, Tel-Aviv University, Tel Aviv, Israel
17  Institute of Ophthalmology, University College London (UCL), London, UK
    *Corresponding author. Tel: +44 207 882 6235; Fax: +44 207 882 6197; E-mail: l.dunkel@qmul.ac.uk
    †These authors contributed equally to this work
    ‡Correction added on 1 June 2016: In the last sentence of the abstract the word "casual" was corrected to "causal".

populations, it is a highly heritable trait, suggesting strong genetic determinants (Wehkalampi *et al*, 2008b). Previous epidemiological studies estimate that 60–80% of the variation in pubertal onset is under genetic regulation (Parent *et al*, 2003; Gajdos *et al*, 2009; Morris *et al*, 2011). However, despite this strong heritability, little is known about the genetic control of human puberty (Palmert & Dunkel, 2012).

Abnormal pubertal timing affects up to 5% of adolescents and is associated with adverse health and psychosocial outcomes (He *et al*, 2010; Ritte *et al*, 2012; Widen *et al*, 2012; Day *et al*, 2015). Our lack of understanding of the factors that trigger pubertal onset is a barrier both to diagnosis and to the management of patients with pubertal disorders, and also hampers attempts to comprehend the population-wide trend toward an earlier age of pubertal onset in the developed world (DiVall & Radovick, 2008; Mouritsen *et al*, 2010).

Attempts to identify key genetic regulators of the timing of puberty have ranged from genome-wide association studies of age at menarche (Ong *et al*, 2009; Elks *et al*, 2010) to next-generation sequencing approaches. Together, these studies suggest that pubertal timing in the general population may be controlled by hundreds of genetic regulators, while loss-of-function mutations in one gene can produce the phenotypic features of complete GnRH deficiency. In patients with hypogonadotropic hypogonadism (HH), up to 30 separate genes resulting in severely delayed or absent puberty have been identified (Bianco & Kaiser, 2009; Gajdos *et al*, 2009). These genes control GnRH neuronal migration and differentiation, GnRH secretion, or its downstream pathways (Karges & de Roux, 2005; Beate *et al*, 2012). Evidence for digenic inheritance of HH, with synergistic effects of two gene defects together producing a more severe phenotype, has also been established (Pitteloud *et al*, 2007).

At the extreme end of the normal range of pubertal onset, self-limited delayed puberty (DP) is a common condition (Sedlmeyer, 2002a). Self-limited DP is defined as the absence of testicular enlargement in boys or breast development in girls at an age that is 2–2.5 standard deviations (SD) later than the population mean (Palmert & Dunkel, 2012). DP segregates within families, with the majority of families displaying an autosomal dominant pattern of inheritance (Sedlmeyer, 2002b; Wehkalampi *et al*, 2008b). Recently, variants in HH genes have been identified in some cases of self-limited DP (Zhu *et al*, 2015). However, in the majority of patients with DP, the neuroendocrine pathophysiology and its genetic regulation remain unclear. Our large, well-phenotyped cohort with self-limited DP from the relatively homogenous Finnish population provides invaluable familial data with which to investigate this question (Kristiansson *et al*, 2008; Wehkalampi *et al*, 2008b). We hypothesized that such families will be enriched for low-frequency, high- or moderate-effect alleles that are amenable to discovery through exome sequencing.

## Results

### Rare, potentially pathogenic variants in the *IGSF10* gene found in 10 families with DP

Initial whole exome sequencing performed in the 18 most extensive families from our cohort (111 individuals: a total of 76 individuals with DP, male = 53 and female = 23; and 35 controls, male = 13 and female = 22) identified 2,474,145 variants after quality control

(Fig 1). Following filtering through our in-house pipeline to identify rare, predicted deleterious mutations, segregating with trait in an autosomal dominant inheritance pattern in multiple families and with potential biological relevance, 28 top candidate genes were identified. These 28 genes were then put forward for targeted resequencing in a further 42 families from the same cohort (178 individuals with DP and 110 controls, Fig 1), and the filtered results were analyzed by applying statistical thresholds for enrichment of rare, pathogenic variants in our cohort via rare variant burden testing with multiple comparison adjustment (Benjamini *et al*, 2001).

The candidate gene, *immunoglobulin superfamily member 10*, *IGSF10* (ENSG00000152580, gene identification number 285313), was identified after rare variant burden testing (adjusted *P*-value = 0.020) and screening of a further 100 controls from our cohort (Fig 1). Four genes had initially passed the *P* < 0.025 threshold after rare variant burden testing, and potentially pathogenic variants in these genes were further analyzed to determine their presence in controls from our cohort and for segregation with trait (Appendix Table S1 and Fig 1). Following this analysis, *IGSF10* was found to be the most promising candidate, with four potentially pathogenic variants in 10 probands from our cohort. The other 9 of 13 rare and potentially pathogenic variants that had been identified in *IGSF10* from targeted exome sequencing results were discarded in our post-sequencing analysis, as they were present in multiple controls from our cohort.

Four variants in *IGSF10* identified in 31 individuals from 10 families (NM_178822.4: c.467G>T (rs138756085) p.Arg156Leu, NM_178822.4: c.481G>A (rs114161831) p.Glu161Lys, NM_178822.4: c.6791A>G p.Glu2264Gly and NM_178822.4: c.7840G>A (rs112889898) p.Asp2614Asn) were found in ≤ 1 control subject (Table 1, Figs 2A and EV1).

Although three of the four variants were present in public databases, they were highly enriched in our cohort (Table 1). Analysis of self-limited DP families is complicated by the fact that this phenotype represents the tail of a normally distributed trait within the population, so it is anticipated that variants that govern the inheritance of this condition will also be present in the general population at a low level. Indeed, it is expected that up to 5% of the individuals sequenced in population databases will have abnormal pubertal timing, either early or delayed. Thus, the absence of these variants in population databases cannot be used as an exclusion criterion, and instead, a comparison of prevalence of such variants must be made to identify those that are enriched in patients compared to the ethnically matched general population.

All four *IGSF10* variants are heterozygous missense variants predicted to be deleterious, damaging, or possibly damaging by ≥ 3/5 prediction tools (Table 2). All variants affect amino acids that are highly conserved among homologues, as revealed by PhyloP or GERP score, and multiple sequence alignment (Table 2 and Appendix Fig S1).

### Families with *IGSF10* variants display autosomal dominant inheritance and classical self-limited DP

Two N-terminal variants in *IGSF10* (p.Arg156Leu and p.Glu161Lys) were identified in 20 individuals from six families (Figs 2A and 3A). Perfect segregation with the expected autosomal dominant pattern of inheritance was seen in all but one individual

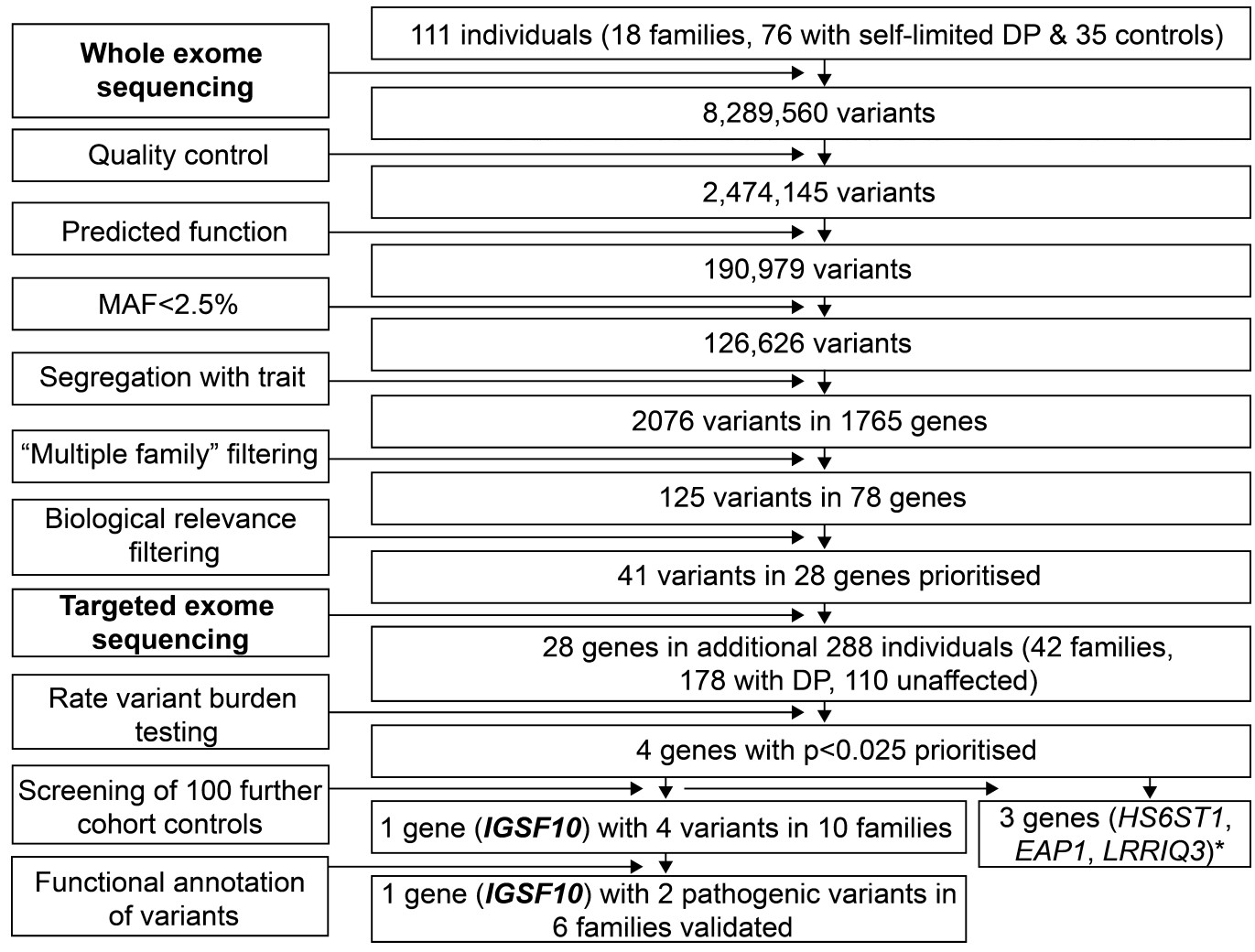

**Figure 1. Flowchart of exome sequencing filtering outcomes.**

Whole exome sequencing was initially performed on DNA extracted from the peripheral blood leukocytes of 111 individuals from the 18 most extensive families from our cohort (76 with DP and 35 controls). The exome sequences were aligned to the UCSC hg19 reference genome. Picard tools and the genome analysis toolkit were used to mark PCR duplicates, realign around indels, recalibrate quality scores, and call variants. Variants were analyzed further and filtered for potential causal variants using filters for quality control, predicted functional annotation, minor allele frequency (MAF), segregation with trait, variants in multiple families, and biological relevance (see Materials and Methods and Appendix Table S1 for further information on filtering criteria). Targeted exome sequencing using a Fluidigm array of 28 candidate genes identified post-filtering was then performed in a further 42 families from the same cohort (288 individuals, 178 with DP and 110 controls). Variants post-targeted resequencing were filtered using the same criteria as the whole exome sequencing data. Rare variant burden testing was performed for all genes selected for targeted resequencing, in order to rank candidate genes post-targeted resequencing. A multiple comparison adjustment was applied to the set of 28 *P*-values *post hoc* (Benjamini *et al*, 2001). Screening of 100 further cohort controls was via conventional Sanger sequencing. Functional annotation of the variants as described elsewhere in Materials and Methods. DP, delayed puberty. *data unpublished.

**Table 1. Minor allele frequency of *IGSF10* variants in study population and control cohorts.**

| Nucleotide change | Amino acid change | Exon | MAF from DP patients (%) (*n* = 215) | MAF from controls (%) (*n* = 210) | MAF (%) Finnish/European/All |
|---|---|---|---|---|---|
| c.467G>T | p.Arg156Leu | 3 | 2.8 | 0 | 0/0.5/0.4 |
| c.481G>A | p.Glu161Lys | 3 | 5.6 | 0.5 | 2.0/0.7/1.0 |
| c.6791A>G | p.Glu2264Gly | 6 | 0.5 | 0 | not seen |
| c.7840G>A | p.Asp2614Asn | 6 | 3.3 | 0 | 0/0.8/0.8 |

Minor allele frequency (MAF) data for the Finnish population were retrieved from The Sequencing Initiative Suomi (The SISu project) (http://www.sisuproject.fi/, release 3.0, accessed September 2015). European and other MAF data were retrieved from the ExAC Browser (Exome Aggregation Consortium (ExAC), Cambridge, MA: http://exac.broadinstitute.org, accessed September 2015).

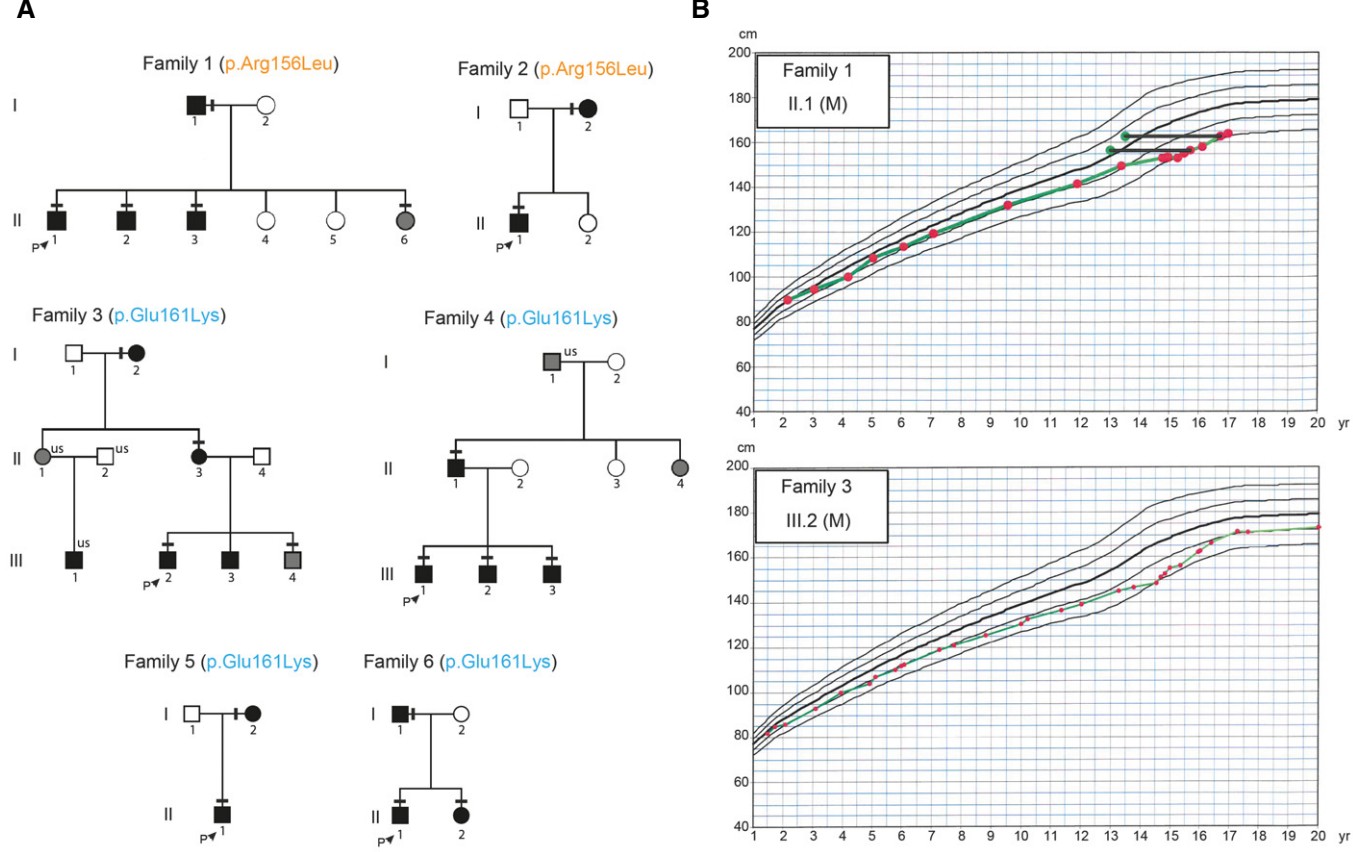

**Figure 2. Pedigrees of the families with N-terminal *IGSF10* mutations with typical growth charts.**

A Squares indicate male family members, and circles female family members. Black symbols represent clinically affected, gray symbols represent unknown phenotype, and clear symbols represent unaffected individuals. The arrow with "P" indicates the proband in each family and "us" indicates unsequenced due to the lack of DNA from that individual. The mutation in each family is given next to the family number; a horizontal black line above an individual's symbol indicates that they are heterozygous for that mutation as identified by either whole exome sequencing (family 3 and 4) or Fluidigm array (family 1, 2, 5, and 6), and verified by Sanger sequencing.

B Growth charts of 2 probands each showing typical growth patterns of self-limited DP, without compromised linear growth before puberty. Solid horizontal black lines connect green dots representing bone age to red dots at the equivalent chronological age.

**Table 2. Prediction of *IGSF10* variants according to web-based prediction software programs and conservation across species.**

| AA Change | dbSNP137 ID | PhyloP (Pollard *et al*, 2010) Pred | SIFT (Kumar *et al*, 2009) Pred | PolyPhen-2 (Adzhubei *et al*, 2010) Pred | LRT (Chun & Fay, 2009) Pred | MutationTaster (Schwarz *et al*, 2014) Pred | FATHMM (Shihab *et al*, 2013) Pred | GERP (Cooper *et al*, 2005) ++ |
|---|---|---|---|---|---|---|---|---|
| p.R156L | rs138756085 | C | D | D | D | N | D | 5.18 |
| p.E161K | rs114161831 | C | D | D | D | D | T | 4.94 |
| p.E2264G | n/a | C | D | P | N | D | T | 3.71 |
| p.D2614N | rs112889898 | C | D | D | D | D | T | 5.24 |

C, conserved; D, deleterious, disease causing or damaging; P, possibly damaging; N, neutral; T, tolerated.

(family3.III.3), who was found to have DP without carrying the variant. Of note given the known association between BMI and pubertal timing, this individual was very lean (weight 13% below median weight for height) at 13 years (Kaplowitz, 2008). The two C-terminal variants (p.Glu2264Gly and p.Asp2614Asn), identified in 11 individuals from four families, in contrast demonstrated

incomplete penetrance in family 7 and a possible *de novo* mutation in family 10 (Fig EV1).

The affected individuals from these 10 families have classical clinical and biochemical features of "simple" DP, with delayed onset of Tanner stage 2 and delayed peak height velocity (Table 3). All probands had low gonadotropins with low or undetectable sex

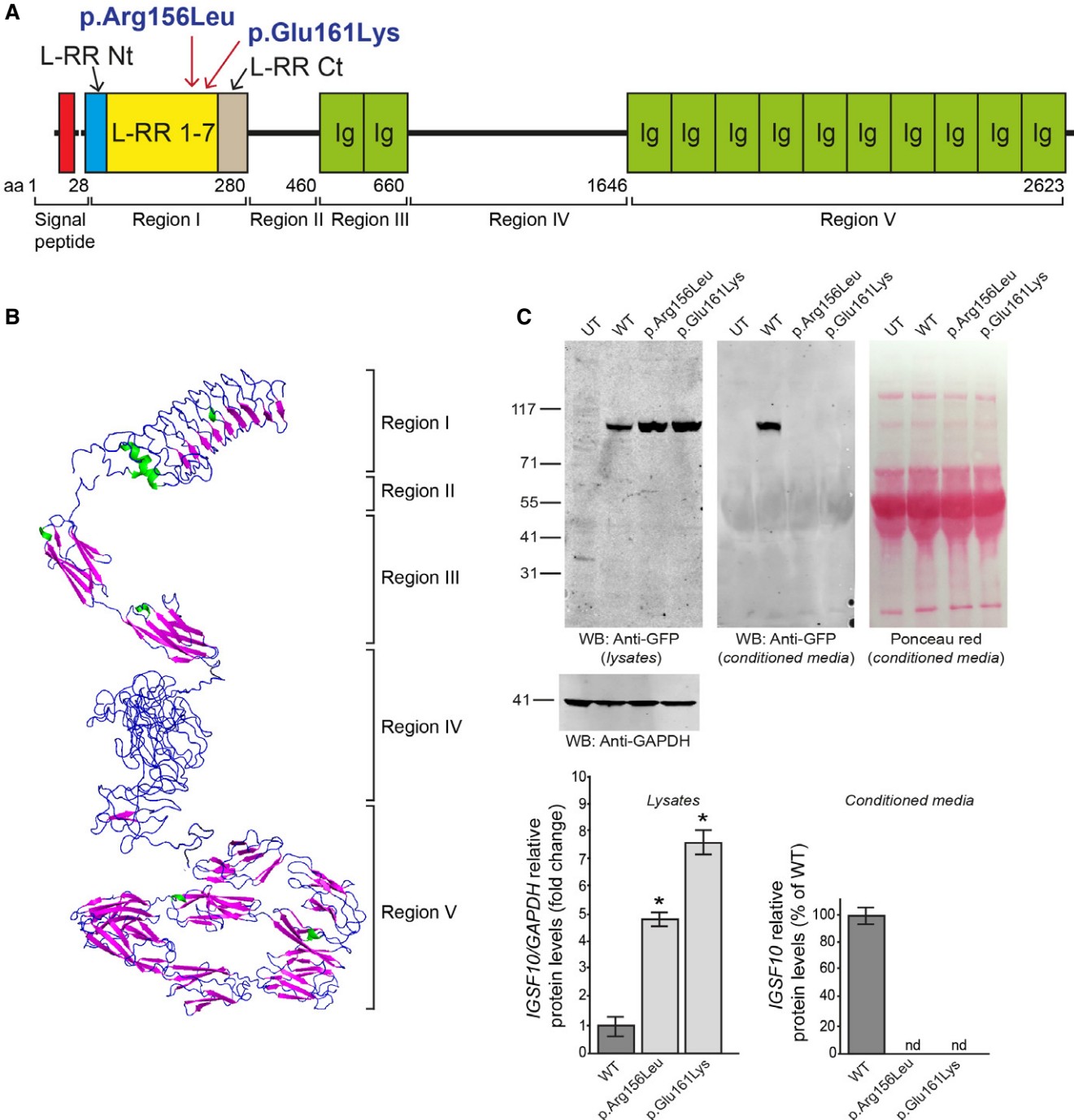

**Figure 3.  IGSF10 protein structure and position of N-terminal mutations.**

A  IGSF10 domains and N-terminal mutations identified in the study. Region I contains leucine-rich repeats (LRR) 1-7 flanked by a LRR N-terminal (LRR Nt) and C-terminal (LRR Ct) cap. Region II is structurally disordered. Region III contains two Ig-like domains (Ig). Region IV is structurally disordered. Region V contains 10 Ig-like domains (Ig).

B  Protein tertiary structure as predicted by *in silico* analysis.

C  Biological consequences of the 2 identified N-terminal mutations. Both WT and mutant N-terminal protein fragments (p.Arg156Leu and p.Glu161Lys) were expressed in HEK293 cells as demonstrated by Western blotting. The GFP-tagged protein products of both were not detected in the conditioned media of mammalian cells, as compared to wild type (WT), and appear to be retained in the intracellular compartment (mean ± SEM; *n* = 3). Ponceau red staining is shown to demonstrate equal protein loading for conditioned media. UT, untransfected negative control; nd, not detected; two-tailed *t*-test, *n* = 3 for each group, *P = 0.01094 (WT vs. p.Arg156Leu) and P = 0.04408 (WT vs. p.Glu161Lys). It has to date not been possible to test cytoplasmic retention for the two C-terminal mutations due to difficulty expressing the full-length or C-terminal protein fragment in mammalian cells.

Source data are available online for this figure.

**Table 3. Clinical and laboratory data of DP probands from each of the 10 families with potentially pathogenic mutations in *IGSF10*.**

| Case | 1.II.1 | 2.II.1 | 3.III.2 | 4.III.1 | 5.II.1 | 6.II.1 | 7.III.5 | 8.III.2 | 9.II.4 | 10.II.1 |
|---|---|---|---|---|---|---|---|---|---|---|
| Clinical data at 1st assessment: | | | | | | | | | | |
| Sex | M | M | M | M | M | M | M | F | M | M |
| Age (years) | 14.76 | 15.5 | 15.5 | 16.01 | 15.69 | 13.66 | 16.14 | 12.18 | 13.55 | 14.94 |
| Bone age[a] | 12.5 | 12.5 | 13.5 | 13.5 | 13.0 | 12.5 | 13 | 10.0 | – | – |
| Tvol | 2.0/2.0 | 2.0/2.0 | 4.0/4.0 | 2.0/2.0 | 2.0/2.0 | 2.0/2.0 | 2.0/2.0 | | 2.0/2.0 | 2.0/2.0 |
| Ph. G or B stage | 1.1 | 1.1 | 1.2 | 2.1 | 1.1 | 1.1 | 1.1 | 1.1 | 1.1 | 1.1 |
| BMI | 19.1 | 18 | 19.4 | 24.9 | 17.9 | 22.8 | 18.6 | 14.5 | 14.5 | 15 |
| LH[b] (IU/l) (0.1–0.6)[c] | 0.3 | 0.1 | 0.4 | 0.1 | 0.2 | 0.1 | 0.1 | 0.1 | 0.15 | 0.15 |
| FSH[b] (IU/l) (0.1–0.9)[c] | 0.45 | 0.1 | 0.5 | 0.3 | 0.6 | 0.35 | 0.6 | 0.2 | 0.5 | 0.3 |
| Testosterone (nmol/l) (0.1–1.0)[c] | 0.3 | 0.22 | 0.4 | 0.55 | 0.3 | 0.22 | 0.2 | – | 0.4 | 0.4 |
| estradiol (pg/ml) (< 8)[c] | – | – | – | – | | – | – | < 5 | | |
| Inhibin B (pg/ml) (55–255)[c] | – | 144 | 121 | 98 | 112 | – | – | | 168 | 155 |
| Age at: | | | | | | | | | | |
| G2 or B2 | 15.2 | 15.6 | 15.5 | 16.5 | 16.10 | 15.21 | 16.5 | 13.94 | 15.21 | 15.4 |
| Takeoff | 14.76 | 15.81 | 15.6 | 16.11 | 15.59 | 15.6 | 16.5 | 13.94 | 15.5 | 15.91 |
| PHV | 15.6 | 16.2 | 17 | 17.3 | 16.2 | 16.18 | 17.3 | 14.68 | 16.1 | 16.4 |
| Induction of puberty | | | | | | | | | | |
| | Yes | No | No | Yes | Yes | Yes | Yes | No | No | No |
| Age at start | 14.76 | | | 16.11 | 15.59 | 15.21 | 16.5 | | | |
| Duration (months) | 3 | | | 6 | 9 | 9 | 6 | | | |
| Olfaction | | | | | | | | | | |
| | Self-reported normal | Self-reported normal | Self-reported normal | Self-reported normal | Self-reported normal | Self-reported normal | Self-reported normal | Self-reported normal | Self-reported normal | Self-reported normal |

Tvol, testicular volume in mls; Ph, pubic hair; G, genital; B, breast stage; IU, international units; G2, genital stage 2; B2, breast stage 2; PHV, peak height velocity.
[a]Bone age estimated by the Greulich and Pyle method.
[b]Baseline values.
[c]Normal ranges for prepubertal boys given in parentheses. Induction of puberty where indicated was with intramuscular testosterone esters.

steroids and delayed bone age at presentation. In addition, these 10 probands displayed a typical growth pattern of self-limited DP, without compromised linear growth before puberty (Fig 2B). Although mean height SDS was below target height, the majority of patients fell within normal limits. At adult height, all but two probands (3.III.2 and 6.II.1) fell within normal limits for distance to target height (Table 4). Birth length, birth weight, timing of pubertal onset, and adult height of those with *IGSF10* mutations were similar to those of other affected DP individuals without *IGSF10* mutations from our cohort (Appendix Table S2).

### *In silico* analysis of *IGSF10*

*IGSF10* is a gene with thus far unclear function, mutations in which have not previously been associated with human disease. The

IGSF10 protein has not previously been modeled by crystallography, so we therefore performed *in silico* analysis (Figs EV2–EV4 and EV5C, and Appendix Fig S2). *In silico* analysis of the protein reveals five defined regions (Fig 3A and B). Region I, in which the two N-terminal variants identified are located, contains leucine-rich repeats (LRR) 1–7 flanked by a LRR N-terminal (LRR Nt) and C-terminal (LRR Ct) cap. Region II is structurally disordered. Region III contains two immunoglobulin-like beta sandwich (Ig-like) domains. Region IV is structurally disordered. Region V, in which the two C-terminal variants are located, contains 10 further Ig-like domains (Fig EV5B and C). No clear evidence for a predicted transmembrane domain was provided by the *in silico* analysis. However, published data give evidence for a putative cleavage site with secretion of the N-terminal portion (Segev *et al*, 2004).

**Table 4.  Growth data of probands with *IGSF10* variants.**

| Case | Sex | Amino acid alteration | Height SDS at the age of 4 years | Height SDS at the age of 8/9 years | Target height | delta HSDS | Distance to target height at 4 years | Distance to target height at 8/9 years | Adult height SDS |
|---|---|---|---|---|---|---|---|---|---|
| 1.II.1 | M | p.Arg156Leu | −1.1 | −1.3 | −0.4 | −0.2 | 0.7 | 0.9 | −0.9 |
| 2.II.1 | M | p.Arg156Leu | −0.2 | −0.4 | −0.7 | −0.2 | −0.5 | −0.3 | 0.1 |
| 3.III.2 | M | p.Glu161Lys | −0.5 | −0.8 | 0.7 | −0.3 | 1.2 | 1.5 | −0.9 |
| 4.III.1 | M | p.Glu161Lys | −0.4 | −0.7 | 0.4 | −0.3 | 0.8 | 1.1 | 0.0 |
| 5.II.1 | M | p.Glu161Lys | −0.3 | −0.2 | 0.9 | 0.1 | 1.2 | 1.1 | 0.9 |
| 6.II.1 | M | p.Glu161Lys | −0.2 | 0 | 1.7 | 0.2 | 1.9 | 1.7 | 0.0 |
| 7.III.5 | M | p.Asp2614Asn | −1 | −1.5 | −0.4 | −0.5 | 0.6 | 1.1 | −0.2 |
| 8.III.2 | F | p.Asp2614Asn | −1.9 | −2.2 | −1 | −0.3 | 0.9 | 1.2 | −1.9 |
| 9.II.4 | M | p.Asp2614Asn | −0.7 | −1.6 | −0.2 | −0.9 | 0.5 | 1.4 | −0.4 |
| 10.II.1 | M | p.Glu2264Gly | −1.8 | −2 | −0.6 | −0.2 | 1.2 | 1.4 | −1.8 |

Height is expressed in SD score (SDS) for national reference data for Finland at 4 years of age, at either 8 years for girls or 9 years for boys, and at adult height. Normal limits: delta HSDS < 1.21, distance to target height at 4 years < 1.76, distance to target height at 8/9 years < 1.72, distance to target height at adult height < 1.44 (Saari *et al*, 2015).

### IGSF10 N-terminal mutant proteins display pathogenic features with failure of extracellular secretion

The two N-terminal variants identified (p.Arg156Leu and p.Glu161Lys) are located in region I within LRR domains (Fig 2A). Both WT and mutant N-terminal protein fragments (668 aa in length) were expressed in HEK293 cells as demonstrated by Western blotting. However, while the GFP-tagged WT protein was detected in the conditioned media, neither mutant protein could be detected in their respective conditioned media. Moreover, a significant increase in mutant protein was detected in cell lysates, suggesting intracellular retention of these two mutants (Fig 3C).

### Tissue expression studies localized Igsf10 mRNA expression to the spatial and temporal window of GnRH neuronal migration

The expression of *Igsf10* mRNA in the nasal region of mouse embryos was analyzed from embryonic day (E) 10.5 to E17.5. During this developmental window, GnRH neurons emerge from the nasal placode (around E11) and then migrate toward the basal forebrain and hypothalamus (E12.5–E17.5) (Cariboni *et al*, 2007). *Igsf10* mRNA expression was undetectable in the nasal region or forebrain at E10.5 (Fig 4A). At E12.5, *Igsf10* was prominently expressed in the nasal mesenchyme (NM) with a decreasing gradient of expression from the area surrounding the vomeronasal organ (VNO) toward the nasal forebrain junction (NFJ), and absent in the VNO and olfactory epithelium (OE) (Fig 4B). At E12.5, GnRH neurons are exiting the VNO and migrating into an *Igsf10* strongly positive cell milieu in the NM (Fig 4B and C). At E14.5, GnRH cells and peripherin-positive olfactory axons are navigating among the *Igsf10*-positive NM cells (Fig 4D, G and H). *Cxcl12*, one of a plethora of molecules known to provide directional cues to GnRH neurons (Memi *et al*, 2013), is also expressed in the NM, although with an opposite gradient compared to *Igsf10* (Fig 4I). At E17.5, GnRH neurons are mainly located in the medial preoptic area (MPOA, Fig 4J). *Igsf10* signal was not detected in the hypothalamus at E17.5. *Igsf10*-positive cells were negative for isolectin b4 (marker of vasculature) at all

developmental stages (Fig 4E, at E14.5). Incubation with the sense probe resulted in no signal at all stages. A similar expression pattern of *IGSF10* was detected in the nasal area of human embryos at 11 post-conceptual weeks (pcw) (Fig 4K–O), with GnRH neurons navigating among the *IGSF10*-positive NM cells.

### Igsf10 knockdown *in vitro* leads to reduced migration of immature GnRH neurons

To investigate the functional role of *Igsf10* in the migration of GnRH neurons, we utilized a model of immortalized but migrating mouse GnRH cells (Radovick *et al*, 1991). These GN11 cells express neuronal markers and retain many features of immature GnRH-secreting neurons (Wetsel, 1995; Gore & Roberts, 1997), including a strong chemomigratory response *in vitro* (Maggi *et al*, 2000; Giacobini *et al*, 2002; Pimpinelli *et al*, 2003). We performed co-culture experiments of GN11 aggregates placed on confluent NIH3T3 monolayers. NIH3T3 cells, derived from a mouse embryonic fibroblast cell line, express high levels of endogenous *Igsf10*. The NIH3T3 cells were treated with scrambled- or Igsf10-shRNAs, the latter leading to highly reduced levels of *Igsf10* expression (Fig 5A). Migration of GN11 across Igsf10-shRNA-treated NIH3T3 cells was found to be significantly reduced compared to that across scrambled shRNA-treated NIH3T3 (two-tailed *t*-test, $n = 3$ for both Igsf10-shRNA and scr-shRNA with at least eight micromass replicates in each group, $P = 0.02422$) (Fig 5B and C, analysis in D).

### Igsf10 knockdown *in vivo* results in perturbed migration and failed neurite extension of GnRH3 neurons

A transgenic zebrafish line, *Tg(gnrh3:EGFP)*, was used to visualize GnRH3 neurons and their projections. RT–PCR analysis indicated high and relatively equal expression of *Igsf10* mRNA at all time-points tested during embryogenesis, from 48 h post-fertilization (hpf) onward (Fig 5E). Morpholino-modified antisense oligonucleotides (MOs) were used to repress *Igsf10* mRNA to assess the effect of *Igsf10* knockdown on the development of the GnRH3

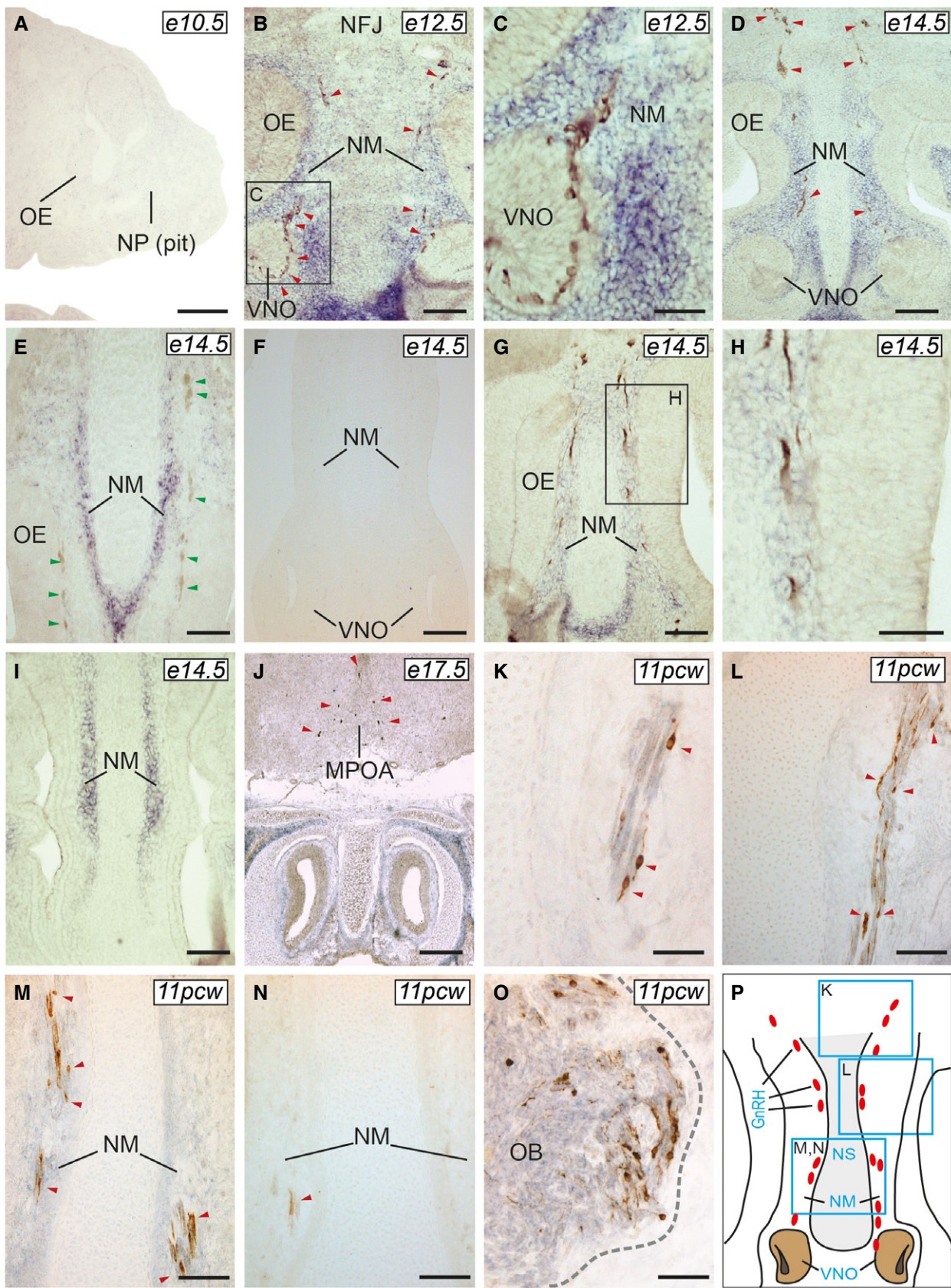

**Figure 4.**

**Figure 4. Expression pattern of *Igsf10* mRNA in mouse and human developing brain.**

A–J    *Igsf10* expression was not observed at E10.5 (A, sagittal section), but readily detectable at E12.5, in the NM (B and C, frontal sections), and E14.5, along the migratory path of GnRH neurons (D, frontal section). *Igsf10*-positive cells were negative for isolectin b4 (a marker of vasculature) at all developmental stages (E, at E14.5). Incubation with the sense probe resulted in no signal at all stages (E14.5 shown in F). Peripherin-positive cells olfactory axons were extending within the *Igsf10*-positive NM (G and H). *Cxcl12* mRNA expression is shown in I (E14.5, frontal section). GnRH neurons are located in the MPOA by E17.5 (J, frontal section) in an *Igsf10*-negative area.

K–P    In human 11pcw brains, *IGSF10* expression pattern was similar to that observed in mouse, with GnRH neurons interspersed in an *IGSF10*-positive NM (K–M, frontal sections, see also schematic in P). *IGSF10*-positive cells were also detected in the olfactory bulb (OB) (O). Sense probe resulted in no specific signal (N).

Data information: GnRH neurons are shown by red arrowheads and isolectin b4 by green arrowheads. NS: nasal septum. Scale bar, 250 μm (A), 100 μm (B, E, H), 50 μm (C, K–O), 200 μm (D, F, G, I, J).

system. Time-lapse analysis of MO-injected *Tg(gnrh3:EGFP)* embryos is shown in Fig 5F. At 48 hpf, GnRH3 neurons are normally seen as bilateral dots in the olfactory organ–olfactory bulb boundary. Over the following days, their projections extend caudally through the telencephalon to the hypothalamus. The strongest effect of morpholino injections was observed at 48 h. At this time-point, the percentage of embryos showing an abnormal GnRH3-neuron phenotype was higher in *Igsf10* splice-site-MO (Sp-MO)-injected embryos compared to relative controls, either injected with a mispair morpholino (control-MO) or uninjected (mean ± standard error of mean: Sp-MO 33% ± 3.6 vs. control-MO 11.7% ± 1.5 vs. uninjected 2.8 ± 1.3; one-way ANOVA, $n = 201/160/156$ for Sp-MO/Ctrl-MO/uninjected, $P = 0.000009$, Fig 5G). A similar effect was observed with the embryos injected with *Igsf10* ATG-MO. *Igsf10* knockdown affected both the guidance and the axonal outgrowth of GnRH3 neurons, which were unable to form compact clusters or extend projections to the hypothalamus.

### Loss-of-function mutation in *IGSF10* in two patients with functional hypogonadotropic hypogonadism

To explore the possible role of mutations in *IGSF10* in conditions of GnRH deficiency, we carried out targeted exome sequencing of *IGSF10* in an adult cohort of 334 patients with HH due to Kallmann syndrome, idiopathic HH, or functional hypogonadism (hypothalamic amenorrhea (HA) or HA equivalent). This investigation showed that 10.2% of these patients carried a rare, predicted damaging variant in *IGSF10* (Appendix Table S3). Three loss-of-function variants (NM_178822: c.C352T: (rs142596318) p.R118*, NM_178822: c.G4804T: (rs79363433) p.E1602*, and NM_178822: c.7353_7354insATCA: (rs570110855) p.L2452 fs) were identified in a total of five patients, and 13 missense variants were identified in 29 patients from our HH cohort, all in the heterozygous state. In particular, all three of the loss-of-function variants were enriched in our HH cohort as compared to the ExAC database.

Two patients (out of 14 patients from this HH cohort with functional hypogonadism) were identified as heterozygous for a shared loss-of-function mutation in *IGSF10* (NM_178822: c.7353_7354insATCA: (rs570110855) p.L2452 fs). This variant is predicted to be deleterious with a high degree of confidence, with the expected loss of the last two Ig-like domains of the IGSF10 protein. It is a rare variant in the general population, with a minor allele frequency in the ExAC database of 0.01%. Both patients carrying the variant had adult-onset functional hypogonadism associated with environmental stressors. Patient 1 aged 31 years had a history of secondary amenorrhea induced by excessive exercise, and presented with failure to achieve spontaneous pregnancy. Patient 2

presented with adult-onset hypogonadism secondary to excessive weight loss and a subclinical eating disorder. Neither patient had a family history of HH, and both had normal pituitary imaging and normosmia. (A more detailed summary is given in Appendix Table S4.)

## Discussion

The genetic control of the timing of puberty remains a fascinating and largely unsolved puzzle. The inheritance of DP is known to be under strong genetic influence with clear autosomal dominant segregation of the trait within families, and thus represents a useful basis for the investigation of puberty genetics. However, the genes responsible for DP have not been identified, other than in a small number of relatives of patients with HH (Lin *et al*, 2006; Pitteloud *et al*, 2006; Tornberg *et al*, 2011; Vaaralahti *et al*, 2011; Zhu *et al*, 2015). In this study, our findings indicate a role for *IGSF10* in the migration of GnRH neurons and highlight two pathogenic mutations in *IGSF10* as the causal factor for DP in six unrelated families. We have identified an additional two rare variants for unknown significance in four further families.

*IGSF10*, a gene of previously unclear function, is a member of the immunoglobulin superfamily. Loss-of-function mutations in another member of this superfamily, *IGSF1*, were recently identified in patients with X-linked central hypothyroidism (Sun *et al*, 2012). Notably, male patients with *IGSF1* mutations have a late increase in testosterone levels with a delayed pubertal growth spurt.

Our functional work specifies a likely role of *IGSF10* in the early migration of GnRH neurons. The development of the HPG axis is exceptional in that GnRH neurosecretory neurons develop in metazoan embryos outside of the central nervous system. Immature GnRH precursor neurons are first detectable in the olfactory placode in the nose from an early embryological stage (E11 in mice) and then begin a complex migration through the forebrain into the hypothalamus and preoptic areas (Cariboni *et al*, 2007; Wray, 2010).

An intact GnRH neurosecretory network is necessary for the correct temporal pacing of puberty, as demonstrated by animal models and the absence of pubertal development in patients with HH (Ojeda *et al*, 2006; Colledge *et al*, 2010; Palmert & Dunkel, 2012). Our tissue expression studies localized *IGSF10* mRNA expression to a spatial and temporal window when GnRH neurons are migrating through the nasal mesenchyme (E11.5–17.5 in mice) to the border with the telencephalon. Further evidence for the role of *IGSF10* in the regulation of GnRH neuronal migration was gained from our *in vitro* demonstration of reduced migration of immortalized GnRH neurons into a cell milieu with strongly reduced *Igsf10*

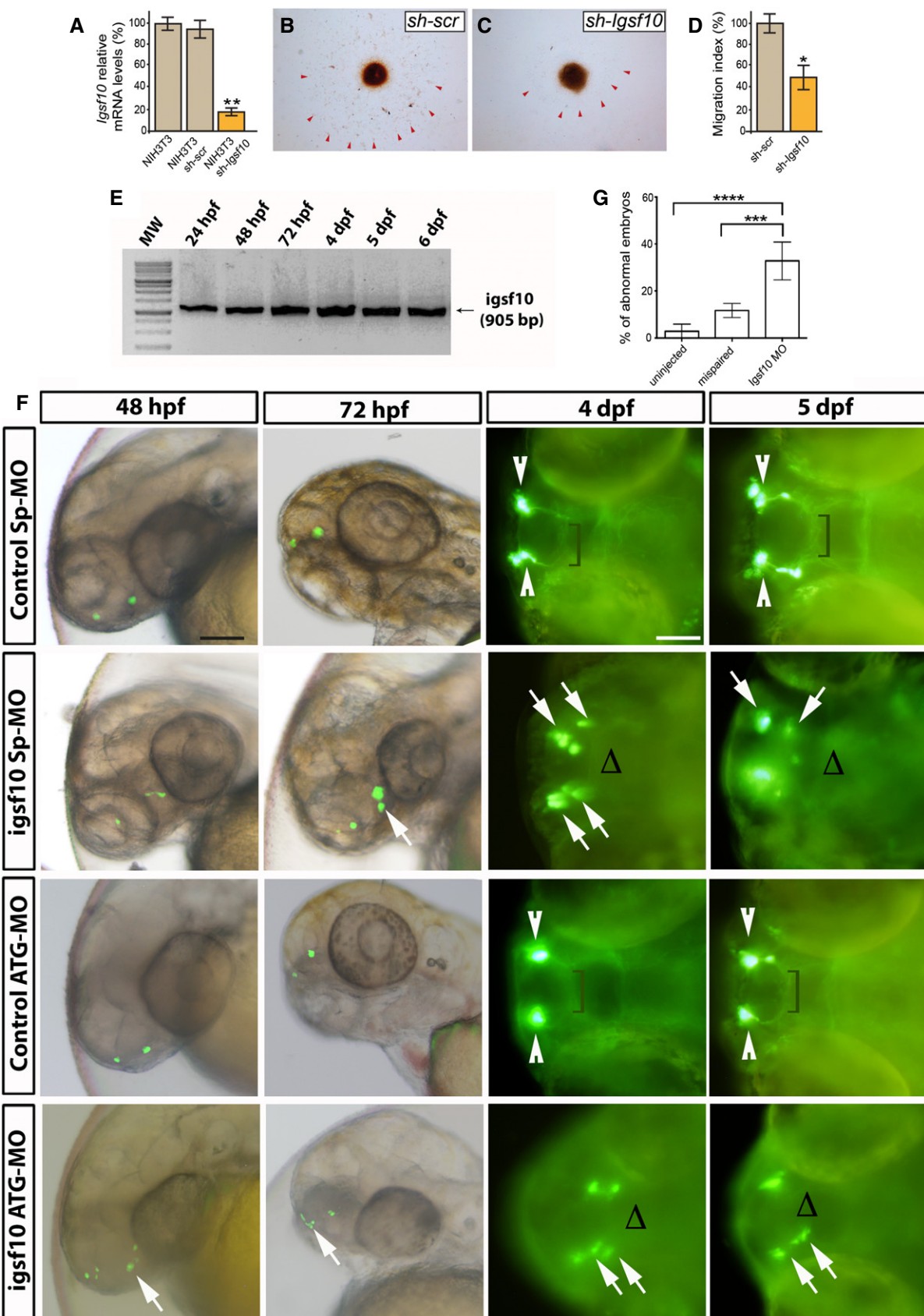

**Figure 5.**

**Figure 5.    Effect of *Igsf10* knockdown on GnRH neuronal migration.**

A       Levels of *Igsf10* expression in native NIH3T3 cells, and cells stably expressing a scrambled (sh-scr) or *Igsf10* (sh-Igsf10) shRNA.

B, C    Migration of GN11 cells from aggregates into NIH3T3 sh-scr (B) or sh-Igsf10 (C).

D       Analysis of the migration index (see Materials and Methods).

E       RT–PCR analysis in zebrafish of total embryos and larvae showing the expression of *Igsf10* mRNA at different time-points. Expected band size: 905 bp.

F       Representative time-lapse analysis of mispair control (control Sp-MO and control ATG-MO) and *Igsf10* morpholinos (Igsf10 Sp-MO and Igsf10 ATG-MO)-injected *Tg (gnrh3:EGFP)* embryos and larvae, at different time-points. Images show lateral view of live larvae head (48 and 72 hpf) and dorsal view (4 and 5 dpf) of live larvae head; anterior is left. White arrowheads indicate normal GnRH3 neuron clusters in the olfactory area of control embryos. Brackets show the extension of the projections toward the hypothalamus. White arrows indicate examples of abnormal GnRH3 neurons scattered in the olfactory area of morphants accompanied by the lack of projections toward the hypothalamus, indicated with Δ (hpf, hours post-fertilization; dpf, days post-fertilization). Scale bar, 100 μm (panels at 48 and 72hpf in F), 150 μm (panels at 4 and 5 dpf in F).

G       Effect of *Igsf10* splice-site morpholino injection observed at 48 h. Quantification of the percentage of embryos showing an abnormal GnRH3-neuron phenotype was significantly higher in *Igsf10* splice-site-MO (Sp-MO)-injected embryos compared to relative controls.

Data information: (A, D) two-tailed *t*-test, *n* = 3 for each group *$P$ = 0.02422, **$P$ = 0.00101. (G) one-way ANOVA with Dunnett's *post hoc* test, *n* = 201/160/156 for Sp-MO/Ctrl-MO/uninjected ***$P$ = 0.00020, ****$P$ = 0.00001. (A, D, G) error bars represent mean ± SEM.

expression. These results were additionally validated by the use of a transgenic zebrafish model, where the depletion of *Igsf10* via injected MOs in zebrafish embryos resulted in perturbed migration and failed neurite extension of GnRH3 neurons toward the hypothalamus.

We hypothesize that immature GnRH neurons may respond to *IGSF10* signaling at the earliest stages of their migration, on exiting the VNO. Pathogenic *IGSF10* mutations, such as those identified here, will disrupt *IGSF10* signaling, potentially resulting in reduced numbers or mis-timed arrival of GnRH neurons at the hypothalamus, the latter leading to a functional defect in the formation of the GnRH neuroendocrine network. With this impaired GnRH system, there would follow an increased "threshold" for the onset of puberty, with an ensuing delay in pubertal timing. This relationship has also been demonstrated in adult Reeler mice, which have significantly fewer GnRH neurons in the hypothalamus and display a phenotype of delayed pubertal maturation (Cariboni *et al*, 2005).

GnRH neurons are known to have receptors for at least 20 neurotransmitters. Migratory GnRH neurons receive a plethora of guidance and movement-inducing messages during this journey, which are likely to be distinct depending on the stage of their migration (Tobet & Schwarting, 2006). Signals may act directly or indirectly through the extension of olfactory axons, as disruption of the nerve tract "scaffolds" themselves can disrupt GnRH migration (Gao *et al*, 2000). Gradients of chemokines may be particularly important for promoting the movement of GnRH neurons (Tobet & Schwarting, 2006). The pattern of *Igsf10* expression in the nasal mesenchyme shows a ventral to dorsal gradient between the VNO and olfactory bulbs, similar to a known axon-guidance gene, *Cxcl12*. *Cxcl12* and its receptor *Cxcr4* are necessary for the guidance of GnRH neurons toward the forebrain in both mouse (Toba *et al*, 2008) and zebrafish (Palevitch *et al*, 2010) models. In the absence of intact Cxcl12/Cxcr4 signaling, GnRH neurons fail to properly migrate out of the nasal mesenchyme during embryonic development.

The discovery of the role of *IGSF10* contributes an additional component to the highly complex system of secreted molecules and chemotactic gradients directing GnRH neuronal migration in the nasal region (Wray, 2010). The specific receptor(s) for secreted IGSF10 protein, on the GnRH neurons or elsewhere, remains to be determined. *IGSF10*, like *GnRH*, is also expressed in lung and other tissues, where its function is as yet unknown. A degree of redundancy with or compensation by other chemokines, such as has been postulated for the GnRH neuron migratory cue semaphorin-4D, as

well as the action of potential "protective factors", is feasible (Wray, 2010).

Furthermore, mutations in *IGSF10* may contribute to the phenotype of other forms of secondary hypogonadism. Mutations that perturb GnRH neuronal migration, including in *KAL1* and *PROKR2*, are already known to cause both HH and HA (Caronia *et al*, 2011). Moreover, the same authors found that 25% of patients with HA in their study had DP. DP is commonly found in relatives of patients with HH and Kallmann syndrome. These findings imply that disruption to the GnRH network may result in a spectrum of phenotypes from DP through HA to complete GnRH deficiency. Specifically, this clinical variability can result from mutations in genes such as *KAL1*, *PROKR2*, and now *IGSF10*, which may lead to late arrival or reduced numbers of GnRH neurons to the hypothalamus, thus compromising the function of the GnRH network. Moreover, an increasing burden of mutations may produce a more severe outcome, with perhaps one mutation leading to DP or HA, while two or even more mutations may be required to lead to a phenotype of HH or KS. Our finding of loss-of-function mutations in *IGSF10* in two out of 14 individuals sequenced with HA and HA equivalent adds weight to this suggestion, although sequencing of *IGSF10* in a larger cohort of HA patients would help to confirm this assertion. However, although rare, predicted pathogenic variants in *IGSF10* were enriched in our HH cohort as compared to the general population, the presence of such variants in the public databases suggests that these variants alone are not sufficient to cause complete idiopathic HH or KS, again in keeping with the above hypothesis.

In conclusion, our findings strongly support the contention that mutations in *IGSF10* cause delayed puberty in humans, through dysregulation of GnRH neuronal migration during embryonic development. Moreover, such mutations may also underlie susceptibility to the phenotype of hypothalamic amenorrhea. Overall, this represents a new causal mechanism for self-limited DP and reveals a shared pathophysiology between DP and other forms of functional hypogonadism.

# Materials and Methods

## Patients

Initial whole exome sequencing was performed in 18 probands with self-limited DP and their affected and unaffected family members

from our previously described cohort (Wehkalampi *et al*, 2008a). In brief, patients referred with DP to specialist pediatric care in central and southern Finland between 1982 and 2004 were identified. All patients ($n = 492$) met the diagnostic criteria for self-limited DP, defined as the onset of Tanner genital stage II (testicular volume > 3 ml) >13.5 years in boys or Tanner breast stage II > 13.0 years in girls (i.e., two SD later than average pubertal development) (Palmert & Dunkel, 2012). Medical history, clinical examination, and routine laboratory tests were reviewed to exclude those with chronic illness. HH, if suspected, was excluded by spontaneous pubertal development at the follow-up. In the 50% of patients from the cohort who choose to have pubertal induction via the use of exogenous sex steroids, all patients were followed up once off treatment until the point of full pubertal development (Tanner stage G4+ or B4+) to ensure that pubertal development did not arrest off treatment.

Families of the DP patients were invited to participate, with information about medical history and pubertal timing obtained by structured interviews and from archived height measurement records. The criteria for the diagnosis of DP in family members were one of the following three: (i) age at takeoff, or (ii) peak height velocity (phv) occurring 1.5 SD beyond the mean, that is, age at takeoff exceeding 12.9 and 11.3 years, or age at phv exceeding 14.8 and 12.8 years in males and females, or (iii) age at attaining adult height more than 18 or 16 years, in males and females, respectively (Wehkalampi *et al*, 2008a). All family members were assigned a clinical status of affected, unaffected, or unknown. Those with unknown status were either too young to diagnose or had insufficient growth data available.

Written informed consent was obtained from all participants.

A hypogonadotropic hypogonadism (HH) cohort of patients with idiopathic hypogonadotropic hypogonadism (IHH, $n = 158$), Kallmann syndrome (KS, $n = 162$), and hypothalamic amenorrhea (HA) or HA equivalent ($n = 14$) were collected through three coordinating European centers (Newcastle upon Tyne Hospital, Newcastle, UK; Bicêtre University Hospital, Paris, France; and Robert Debré Hospital, Paris, France). Patients with IHH all had absent or incomplete puberty at 18 years of age, low or normal serum gonadotropin levels, low serum estradiol or testosterone levels, otherwise normal anterior pituitary function, and normal results on neuroimaging. KS patients had absent or incomplete puberty at 18 years of age, low or normal serum gonadotropin levels, low serum estradiol or testosterone levels, otherwise normal anterior pituitary function, and anosmia diagnosed clinically and by radiological evidence of olfactory bulb disruption. Patients with HA or HA equivalent had a history of spontaneous pubertal development with secondary amenorrhea in women for 6 months or more, low or normal gonadotropin levels, low serum estradiol or testosterone levels, and one or more predisposing factors. These factors included excessive exercise (> 5 h per week), or other stress, loss of more than 15% of body weight, and either evidence of a subclinical eating disorder, or another cause of dietary restriction (eg. self-perceived "intolerance" to certain foods). Patients with frank anorexia nervosa or BMI < 17 kg/m$^2$ were excluded.

## DNA sequencing and bioinformatics

Whole exome sequencing was initially performed on DNA extracted from the peripheral blood leukocytes of 111 individuals from the 18

most extensive families from our cohort (a total of 76 individuals with DP: male = 53 and female = 23, and 35 controls, male $n = 13$ and female $n = 22$), with exome capture on a Nimblegen V2 platform or Agilent V5 platform and sequencing on the Illumina Hiseq 2000. The exome sequences were aligned to the UCSC hg19 reference genome. Picard tools and the genome analysis toolkit were used to mark PCR duplicates, realign around indels, recalibrate quality scores, and call variants.

Variants were analyzed further and filtered for potential causal variants using filters for quality control, predicted functional annotation, minor allele frequency, segregation with trait, variants in multiple families, and biological relevance (Fig 1). Quality control included thresholds for call quality, read depth, and upstream pipeline filtering. Predicted functional annotation involved prioritizing nonsense, exonic missense, splice-site variants, structural or promoter changes, or variants deleterious to a microRNA. Filtering by MAF entailed including those variants with minor allele frequency (MAF) < 2.5% in the 1000 Genomes database, the NHLBI exome variant server, Sequencing Initiative Suomi (sisu.fimm.fi), and dbSNP databases. Segregation with trait refers to variants present in $\geq n-1$ affected individuals (where $n$ = the number of affected individuals in a given pedigree) and not present in more than one unaffected individual being retained. Multiple family filtering involved retaining those variants seen in more than one family from the cohort sequenced, or with different variants in the same gene in more than one family. Biological relevance filtering allowed prioritization of those remaining variants under three criteria: (i) Variants in genes known to be relevant to the phenotype of HH; (ii) variants in genes in linkage disequilibrium ($r^2$ no limit, D prime > 0.8) with loci associated with genome-wide association studies of age at menarche (Elks *et al*, 2010); and (iii) variants in genes with potential biological significance, using the tools Ingenuity Variant Analysis (QIAGEN Redwood City, www.qiagen.com/ingenuity), Genego MetaCore (Thomson Reuters), Ensembl variant effect predictor, and Annovar (Wang *et al*, 2010).

Targeted exome sequencing using a Fluidigm array for library preparation followed by Illumina Miseq sequencing of 28 candidate genes identified post-filtering was then performed on DNA extracted from the peripheral blood leukocytes of a further 42 families from the same cohort (288 individuals, 178 with DP and 110 controls, Fig 1). Variants post-targeted resequencing were filtered using the same criteria as the whole exome sequencing data: on the basis of quality control, predicted functional annotation, minor allele frequency, and segregation with trait.

Rare variant burden testing was performed for all 28 genes posttargeted resequencing. Fisher's exact test was used to compare the prevalence of deleterious variants in our cohort with the Finnish population, using the ExAC Browser (Exome Aggregation Consortium (ExAC), Cambridge MA: http://exac.broadinstitute.org, accessed September 2015). For each of the 28 genes, all variants from the ExAC database with minor allele frequency < 2.5%, predicted to be deleterious by both Polyphen-2 (Adzhubei *et al*, 2010) and SIFT (Kumar *et al*, 2009), were included in the analysis, with each family in our cohort represented by the proband only. A multiple comparison adjustment was applied to the set of 28 *P*-values *post hoc* using the Benjamini and Hochberg method (Benjamini *et al*, 2001). Variants identified following filtering

pipelines were confirmed by conventional Sanger sequencing and screened in a further 100 controls from our cohort via Sanger sequencing.

The HH cohort (*n* = 334) was screened via targeted exome sequencing for mutations in *IGSF10* using a Fluidigm array for library preparation followed by Illumina Miseq sequencing, with the same filtering pipeline used for the whole and previous targeted exome sequencing data. Variants identified were confirmed by conventional Sanger sequencing.

### Growth pattern analysis

The pattern of prepubertal growth in the 10 probands was analyzed using two parameters: (i) HSDS distance from target height (TH) at the ages of 4 and 8 (girls) or 9 (boys) years (TH formula = 0.791 × mean parental height SDS −0.147 for girls and 0.886 × mean parental height SDS −0.071 for boys) and (ii) change in height SDS (ΔHSDS) between the ages of 4 and 8/9 years. Normal values for the two parameters, based on data from > 70,000 healthy Finnish children, have been previously published (Saari *et al*, 2015).

### *In silico* analysis

The amino acid sequence for human *IGSF10* was retrieved from UNIPROT database (id Q6WRI0). Homology modeling was used to determine the 3D structure using two high-performing protein structure prediction servers: Phyre2 (Kelley & Sternberg, 2009; Jefferys *et al*, 2010) and I-Tasser (Roy *et al*, 2010). The signal peptide cleavage site was calculated using SignalP4.1 (Petersen *et al*, 2011). The following interactions involved in protein stability were considered: (i) salt bridges, defined as at least one pair of atoms on oppositely charged groups within a 4.5 Å distance; (ii) hydrogen bonds (H-bond), defined as a donor–acceptor distance ≤ 2.5 Å and an angle at the acceptor ≥ 90°; and (iii) disulfide bridge (S-S bridge) defined as the side chains of two cysteines at a 3.0 Å distance. Pairs of cysteines at a greater distance were also considered as potentially forming an S-S bridge when found to be reasonably close (Cα-Cα distance < 10 Å). We used Cα distance to allow for errors in side-chain placement and a relatively high threshold to accommodate possible deviation of the backbone from native. The protein electrostatic potential was calculated using PBEQ program (Jo *et al*, 2008), which computes the protein electrostatic potential by solving the Poisson–Boltzmann equation. Protein structures were visualized using the Pymol visualization program (http://www.pymol.org/).

### Constructs and protein expression

An N-terminal fragment encoding the first 668 aa of *IGSF10* gene (RefSeq NM_178822.3) was cloned into a pcDNA-EGFP (Addgene plasmid #13031), and p.Arg156Leu and p.Glu161Lys variants were generated using PCR-mediated mutagenesis (Quickchange II, Agilent Technologies) according to the manufacturer's instructions and verified by sequencing.

HEK293 (sourced from ATCC) were transfected with wild-type or mutant plasmids using polyethylenimine (Sigma) via a standard protocol. Cells were checked for mycoplasma contamination (MycoAlert Detection Kit, Lonza) on a monthly basis and were contamination-free. Transfected cells were selected with G418

(1 mg/ml, Sigma). Once stable cell lines were established, the cells were cultured for 24 h in 6-well plates to full confluency in 1 ml of media. This process was repeated on three separate occasions with cells from the same stable cell lines, to produce technical replicates. Conditioned media from each well were removed for analysis before cell lysis with RIPA buffer (Sigma) supplemented with protease inhibitors (Complete Mini, Roche). Lysates and conditioned media were cleared by centrifugation at 16,060 *g* at 4°C for 10 min. Equal amounts of lysates and conditioned media were size-separated (NuPage BisTris gels 4–12%) and transferred to nitrocellulose membranes (Promega). Post-transfer, the blots were stained with Ponceau red to assess equal loading, blocked with PBS containing 0.1% Tween-20 and 5% nonfat dry milk, and incubated in a 1:500 dilution of mouse monoclonal anti-GFP (Roche, 1DB-001-0000570956) antibody or a 1:3000 dilution of mouse monoclonal anti-glyceraldehyde-3-phosphate dehydrogenase (GAPDH, Santa Cruz Biotechnology, 1DB-001-0000183498) for 12 h at 4°C. After washes, the membranes were incubated with goat anti-mouse IRDye680 (1:10000 dilution; Licor). Immunoblots were scanned with, and protein relative amounts were calculated by, the Odyssey® Fc Imaging System (Licor).

### Nonradioactive *in situ* hybridization (NR-ISH)/Immunohistochemistry

E10.5 to E17.5 mouse embryos were collected from timed crosses of C57BL/6 mice. The morning of the vaginal plug was designated 0.5 days; 11 post-conceptual weeks (pcw) human brains were obtained from the MRC-Wellcome Human Development Biology Resources (HDBR—Institute of Genetics Medicine, Newcastle, UK). Tissues were fixed in 4% paraformaldehyde (PFA) in PBS, cryoprotected in 30% sucrose, and frozen in OCT compound (VWR); 12-μm-thick serial sagittal and coronal sections were collected on Superfrost Plus slides (VWR).

Mouse and human *Igsf10* were PCR-amplified from brain cDNAs using the following primers: m*Igsf10* FOR: 5′-GCAAGAAGGAAA-GAATCCCC -3′, REV: 5′- GATTCGCCCATCCTCACTAA -3′; h*IGSF10* FOR: 5′-TCAGGAGCTTGACACGATTG-3′, REV: 5′- CTGCGGTGTTT CACTAAGCA-3′. Amplified cDNAs were cloned into the dual promoter vector pGEM-T easy (Promega) and linearized with the appropriate restriction enzymes. Mouse Cxcl12 probe was from Memi *et al* (2013). Probe preparation and *in situ* protocol were performed as previously in Guasti *et al* (2011).

When co-labeling was desired, after *in situ*, the sections were incubated with primary antibodies (anti-GnRH, Immunostar; anti-peripherin, Merck-Millipore; anti-isolectin B4, Sigma) diluted 1:1000 in PBS–Triton 0.1%, overnight at room temperature (RT) as used in Cariboni *et al* (2011). After three washes with PBS–Triton 0.1%, the slides were incubated for 2 h at RT with biotin-conjugated goat secondary antibodies (Vector Laboratories), diluted 1:300 in PBS and, after further washes, with the avidin–biotin complex (ABC staining kit, Vector Laboratories). The sections were reacted with 4′,6′-diamino-2-phenylindole (DAPI, Vector Laboratories) and mounted as above.

Images were acquired using a Leica DM5500B microscope (Leica, Nussloch, Germany), equipped with a DCF295 camera (Leica) and DCViewer software (Leica), and then processed with Abode Photoshop CS6 and Adobe Illustrator CS6.

## Migration experiments

*Igsf10* silencing was achieved in NIH3T3 cells (sourced from ATCC) using SureSilencing shRNA plasmids (QIAGEN) according to the manufacturer's instructions. Stable lines were obtained by puromycin treatment (Life Technologies, 2 g/ml). Cells were checked for mycoplasma contamination (MycoAlert Detection Kit, Lonza) on a monthly basis and were contamination-free. The level of *Igsf10* knockdown was assessed by RT–qPCR. RNA extraction (RNeasy Mini kit, QIAGEN) and cDNA synthesis (M-MLV Reverse Transcriptase, Promega) were performed according to the manufacturers' instructions. RT–qPCR was performed in a 10-μl reaction mixture containing 2 μl cDNA template, 5 μl 2 × SYBR$_{GREEN}$ I Master Mix (KAPA Biosystems), 0.2 μl low ROX (KAPA Biosystems), 0.5 μl primers (10 μM forward + reverse), and 2.3 μl nuclease-free H$_2$O. *Gapdh* was used as the endogenous housekeeping gene. The real-time PCR was performed using an Mx3000 Thermocycler (Stratagene) using the following primers and conditions: m*Igsf10*: FOR, 5′-CTGGGGAGTCCAATTGCTGT-3′ and REV, 5′-GCTGCCTTTGCTGACATC-3′ (18 bp); *Gapdh*: FOR, 5′-TGCACCACCAACTGCTTAG-3′ and REV, 5′-GGATGCAGGGATGATGTTC-3′. Quantitative RT–qPCR was set up in triplicate. After an initial denaturation step of 3 min at 95°C, PCR cycling was performed for 40 cycles of 95°C for 3 s, 55°C for 20 s and 72°C for 1 s, followed by 1 cycle of 1 min at 95°C, 55°C for 30 s and 95°C for 30 s. Silencing was achieved at 80% in *Igsf10*-silenced NIH3T3 cells compared to cells expressing the scrambled shRNA.

Micromass cultures were obtained with the immortalized GnRH-expressing cell line GN11. Confluent monolayer cultures of GN11 were released by trypsin–EDTA and resuspended in growth medium. Micromasses were obtained by pipetting 20 μl (4.0 × 10$^5$ cells) of cell suspension onto the lid of a 10-cm petri dish and incubating for 48 h in a 5% CO$_2$ incubator at 37°C. Micromasses were then placed onto confluent scrambled and *Igsf10*-silenced NIH3T3 cells for 7 days. The cells were then fixed in 4% PFA, and GN11 migration into NIH3T3 was assessed by staining with anti-GnRH antibodies, as used in Cariboni *et al* (2011). Migration index was calculated by assessing GnRH-stained cells using the ImageJ software (NIH). Statistical significance was evaluated by ANOVA and Student's *t*-test. *$P$ < 0.05 and **$P$ < 0.01 were taken to be significant. Data are presented as mean ± standard error of mean and expressed as percentage reduction of migration, taking untreated or scrambled controls as 100%. Each experiment was performed using at least 12 micromasses/group and repeated three times.

## Zebrafish investigations

### Animals

A transgenic line, *Tg(gnrh3:EGFP)*, was used to visualize GnRH3 neurons and their projections (Abraham *et al*, 2008). Wild-type and transgenic zebrafish embryos were generated by natural mating. Both AB and TL strains were used for the experiments. Embryos were raised in fish water with methylene blue (0.3 p.p.m.) in petri dishes at 28°C in a 12-h light/12-h dark cycle. Developmental stages are expressed in hours post-fertilization (hpf) or days post-fertilization (dpf). All embryos used in experiments were 5 dpf or younger. Pigmentation was prevented by adding 0.2 mM phenylthiourea (PTU) to the water at 24 hpf.

### RT–PCR analysis

Total RNA was isolated from embryos and larvae at different stages of embryogenesis using EZ RNA Total RNA Isolation kit (Biological Industries, Beit Haemek, Israel). First-strand cDNA was synthesized from 1 μg of total RNA by qScript cDNA kit (Quanta Biosciences, Gaithersburg, MD, USA) according to the manufacturer's protocol. Primers for Igsf10: FOR, 5′-TTGGCTACAGTCCCGATTTC-3′ and REV, 5′-AAATTTTGCTGGGACGAATG-3′.

### Morpholino-knockdown experiments

Morpholino-modified antisense oligonucleotides (MOs) (Gene Tools, Philomath, OR, USA) were used to repress *Igsf10* mRNA to assess the effect on the development of the GnRH3 system. A splice-site-MO (Igsf10 Sp-MO 5′-GCCTGTTGTAGGTTTTACCCCAGGT-3′, 25 bp) and an ATG-MO (Igsf10 ATG-MO 5′-GGGAATCCGCTGCTGGGTCA-CACAT-3′, 25 bp) were designed. Quantities of 1.5pmol/embryo (Igsf10 Sp-MO) or 1pmol/embryo (*Igsf10* ATG-MO) were injected into *Tg(gnrh3:EGFP)* embryos immediately after fertilization. In all experiments, Igsf10 Sp-MO- or ATG-MO-injected embryos were compared with embryos injected with the same amount of mispair control (control Sp-MO 5′-GCGTCTTGTACGTTTTACCCGACGT-3′ or control ATG-MO 5′-GAGAATACGCTACTGGGTAACAAAT-3′, both 25 bp) at the same developmental stage.

Two sets of experiments (biological replicates) were performed and each experiment consisted of three groups: Sp-MO injected, Ctrl-MO injected, and uninjected. No statistical method was used to predetermine sample size. Rather, sample size was based on preliminary data and observed effect sizes. Dead embryos or embryos that were GFP negative at 48 hpf were excluded from the analysis. Considering that MO injection can cause a death rate of 10–25% in 24-h-old embryos, we injected a large number of embryos in order to establish a statistical significance. In the first experiment, the mortality rate in the injected embryos, Sp-MO and Ctrl-MO, was 11% and in the second, it was 25%. The mortality rate in the uninjected embryos was 2.5 and 5.7%, respectively.

In the first experiment, 353 embryos were injected with Sp-MO, 260 embryos were injected with Ctrl-MO, and 244 embryos left uninjected: 146 of 353, 108 of 260, and 100 of 244 embryos were available and GFP positive for analysis at 48 hpf. In the second experiment, 186 embryos were injected with Sp-MO, 157 embryos were injected with Ctrl-MO, and 122 embryos left uninjected: 55 of 186, 52 of 157, and 56 of 122 embryos were available and GFP positive for analysis at 48 hpf. Sex of embryos was undetermined (separate sexes can be detected only after 21–23 dpf).

Injected larvae were checked daily and the EGFP expression was monitored under SZX12 fluorescent dissecting microscope (Olympus, Tokyo, Japan) for 5 days. Quantification of the percentage of embryos showing an abnormal GnRH3-neuron phenotype in *Igsf10* splice-site-MO (Sp-MO)-injected embryos compared to relative controls, injected either with a mispair morpholino (control-MO) or uninjected, was assessed by mean ± standard error of mean, tested by one-way ANOVA followed by Dunnett's *post hoc* test.

Efficacy of Igsf10 Sp-MO was evaluated by RT–PCR. Injected and uninjected embryos were sampled at 48 hpf and total RNA was extracted (EZ RNA Total RNA Isolation kit, Biological Industries, Beit Haemek, Israel); 1 μg of RNA was retro-transcribed using qScript cDNA kit (Quanta Biosciences, Gaithersburg, MD, USA). Specific primers corresponding to zebrafish *Igsf10* exons 2 and 3

(Igsf10-E2FOR 5′-GCGGATTCCCGTCTGCTATGG-3′ and Igsf10-E3REV 5′-TGCAGAGATGTGAGGCCACTGAAC-3′) were used for PCR amplification. PCR products were separated on an agarose gel, extracted from gel using HiYield Gel/PCR DNA Fragments Extraction Kit (RBC Bioscience, New Taipei City, Taiwan), and sequenced. Sequencing analysis demonstrated the insertion of 97 nucleotides within the intron 2 of Igsf10 Sp-MO-injected embryos, which would result in a frameshift and produce a dysfunctional Igsf10 protein in these embryos (Appendix Fig S3).

## Study approval

The study protocol was approved by the Ethics Committee for Pediatrics, Adolescent Medicine and Psychiatry, Hospital District of Helsinki and Uusimaa (and extended to encompass Kuopio, Tampere, and Turku University Hospitals). UK ethical approval was granted by the London-Chelsea NRES Committee. The study was conducted in accordance with the guidelines of the Declaration of Helsinki. All patients gave informed written consent prior to inclusion in the study. All mouse experiments were conducted under the Animal (Scientific Procedures) Act 1986, Project Licence PPL 70/8269. The human embryonic and fetal material was provided by the Joint MRC/Wellcome Trust Grant# 099175/Z/12/Z Human Developmental Biology Resource (http://hdbr.org).

## Statistics

For rare variant burden testing, Fisher's exact test was used with a multiple comparison adjustment applied *post hoc* using the Benjamini and Hochberg method (Benjamini *et al*, 2001). A threshold of $P < 0.025$ was taken as significant. Fisher's exact test was also used for the analysis of HH cohort allele frequencies. The unpaired *t*-test (two-tailed) was used for statistical analysis of auxological data (Appendix Table S2).

For all experiments, data are expressed as the mean ± SEM. To determine the statistical significance, we used the unpaired *t*-test (two-tailed) or, for multiple comparisons, a one-way ANOVA followed by a Dunnett's *post hoc* test. A *P*-value of less than 0.05 was considered statistically significant. Statistical analysis was performed using GraphPad Prism4 (GraphPad Software).

Expanded View for this article is available online.

## Acknowledgements
SRH is funded by The Wellcome Trust (102745), Rosetrees Trust (M222), and the Barts and the London Charity (417/1551). LG and GRB are funded by the Biotechnology and Biological Sciences Research Council (BB/L002671/1). LD is partly supported by the Academy of Finland (14135). MRB, HRW, and CPC are funded by the National Institutes for Health Research (NIHR), and this work forms part of the portfolio of translational research of the NIHR Biomedical Research Unit at Barts. AD is funded by the MRC (MR/K021613/1). AC is funded by the Telethon Foundation (GP13142); VA is partly supported by a COST STSM (BM1105-16145) and a Travel Grant sponsored by Developmental Journal (The Company of Biologists Limited). The human embryonic and fetal material was provided by the Joint MRC/Wellcome Trust Grant# 099175/Z/12/Z Human Developmental Biology Resource (http://hdbr.org). We would especially like to thank all patients and their families who participated in the study.

## The paper explained

### Problem
Early or late pubertal onset affects up to 5% of adolescents and is associated with adverse health and psychosocial outcomes. Self-limited delayed puberty segregates in families, suggesting strong genetic influences, but the underlying mechanistic and genetic determinants of the pathogenesis of delayed puberty are not understood.

### Results
Using whole and candidate exome sequencing in a large cohort with familial delayed puberty, we have identified a novel gene, *IGSF10*, mutations in which lead to self-limited delayed puberty. Our functional annotation of *IGSF10* suggests that it is a component of the complex system of migratory cues, which together guide GnRH neurons from their origin in the nasal placode toward the hypothalamus during embryogenesis. Loss-of-function mutations in *IGSF10* were also identified in patients with hypothalamic amenorrhea.

### Impact
Here, we identify a new causal mechanism for self-limited delayed puberty, through defects in embryological migration of GnRH neurons to the hypothalamus, and reveal a shared pathophysiology between simple delayed puberty and other forms of functional hypogonadism.

## Author contributions

SH, LG, and LD planned the genetic and experimental work. SH, LG, AM, and GR-B carried out the *in vitro* analysis. VA and YG carried out the zebrafish work. CC, HW, and MB gave bioinformatics and statistical support. AD and MS carried out *in silico* analyses. LD, KW, RQ, NdR, JY, and AG-M provided clinical samples and data. SH, LG, LD, AC, MB, HS, and LM contributed to data review and interpretation.

## Conflict of interest
All authors declare that they have no conflict of interest.

## For more information
IGSF10
http://www.uniprot.org/uniprot/Q6WRI0
http://www.genecards.org/cgi-bin/carddisp.pl?gene=IGSF10
Delayed puberty
http://patient.info/doctor/delayed-puberty

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
