## [Review Process File · EMBO Molecular Medicine]

IGSF10 mutations dysregulate gonadotropin-releasing hormone neuronal migration resulting in delayed puberty

Sasha R Howard, Leonardo Guasti, Gerard Ruiz-Babot, Alessandra Mancini, Alessia David, Helen L Storr, Lousie A Metherell, Michael JE Sternberg, Claudia P Cabrera, Helen R Warren, Michael R Barnes, Richard Quinton, Nicolas de Roux, Jacques Young, Anne Guiochon-Mantel, Karoliina Wehkalampi, Valentina Andrè, Yoav Gothilf, Anna Cariboni, Leo Dunkel

Corresponding author: Leo Dunkel, William Harvey Research Institute, Barts and the London School of Medicine and Dentistry, Queen Mary University of London

Review timeline:	Transfer date:	26 January 2016
	Editorial Decision:	12 February 2016
	Revision received:	01 March 2016
	Accepted:	14 March 2016

Transaction Report:

Editor: Céline Carret

1st Editorial Decision	12 February 2016
------------------

Thank you for the submission of your manuscript to EMBO Molecular Medicine. We have now heard back from the two referees who we asked to evaluate your manuscript.

You will see that both referees are enthusiastic about the study and only request minimal revision work. I will be happy to invite a revision of your manuscript if you can address the issues that have been raised within 3 months. Please note that it is EMBO Molecular Medicine policy to allow only a single round of revision and that, as acceptance or rejection of the manuscript will depend on another round of review, your responses should be as complete as possible.

In order to gain time, shall the manuscript be later accepted, I would like to suggest taking care of the below editorial requirements at the same time.

I look forward to seeing a revised form of your manuscript as soon as possible.

***** Reviewer's comments *****

Referee #1 (Comments on Novelty/Model System):

The study presents novel findings that will help in the diagnosis of delayed puberty in patients and will contribute to our understanding of the mechanisms governing GnRH migration and development.

Referee #1 (Remarks):

In the present study, Howard and colleagues present a compelling series of human genetic, in vitro and in vivo studies that elegantly describe the novel role of IGSF10 in the migration of GnRH neurons to the hypothalamus during the embryonic period. Their data is supported by a large analysis of patients suffering hypogonadotropic hypogonadism and the function of this molecule postulated and assessed by in vitro models using GN11 cells, which showed reduced migration after Igsf10 knockdown, and reduced migration of GnRH neurons in zebra fish with a morpholino knockdown approach of this molecule. Overall, the study is innovative, informative, well designed and the results clearly stated. There are only a few comments regarding the proposed mechanism of action for Igsf10:

- The authors tested the migration of GN11 cells after KD of Igsf10. This experiment assumes that GnRH neurons express Igsf10, which would be acting, perhaps, in an auto synaptic feedback loop in GnRH neurons. Still, the authors showed the expression of Igsf10 in other hypothalamic areas and, in the discussion, mentioned that this molecule probably participates in the creation of a gradient needed to direct GnRH neuronal migration. It is therefore not clear whether GnRH neurons may also express this molecule or whether GN11 cells, due to their immortalized nature, are not a faithful replication of GnRH neurons in vivo. It would be good if the authors clarified this by assessing the expression of Igsf10 in other GnRH cell lines and, if possible, through double label ISH with better resolution than the images presented in Figure 4.

- The authors nicely explain that the amount of mutations accumulated in a single individual may account for the different magnitudes in the HH phenotype observed, however, this does not explain the adult onset of HH discussed in lines 321+. If the role of Igsf10 is solely in the migration of GnRH neurons, as suggested by the disappearance of its expression in late embryonic phases, why would these mutations induce secondary amenorrhea after the HPG axis has been properly activated during puberty? Do they know whether this molecule has further developmental regulation? Would it be possible that its expression increases again at the time of puberty onset and/or is regulated by sex steroids in adulthood? Including a few samples from mice at critical developmental time points (e.g. infantile, juvenile, peripubertal and adult) would address this question.

- Line 193: do they mean "presence" instead of absence?

- Line 197: One of the mutations has less than 3 D and would be therefore "possibly damaging" according to their criteria.

- Figure 4: The data depicting IGSF10 expression in the human tissue is too weak. Are they sure this is specific? Please, include controls using the sense probe in the supplementals.

- Figure 4: Please, include a scale bar in each panel.

Referee #2 (Comments on Novelty/Model System):

The authors demonstrate that IGSF10 regulates embryonic GnRH neuronal migration and mutations result in delayed puberty. This is, indeed, a new concept in Molecular Medicine. The manuscript has a high technical quality and the information is novel.

Referee #2 (Remarks):

Sasha et al, with the corresponding author being Prof. Dunkel, present an elegant multinational study where they have identified mutations in IGSF10 in 6 unrelated families, resulting in intracellular retention of this protein, thus with failure in the secretion of mutant proteins. Furthermore, the authors show that knock out of IGSF10 caused reduced migration of immature GnRH neurons in vitro and perturbed migration and extension of GnRH neurons in a *gnrh3:EGFP* zebrafish model. Furthermore, loss-of-function mutations in IGSF10 were identified in patients with hypothalamic amenorrhea. The authors conclude by saying that mutations in IGSF10 cause delayed

puberty in humans with common genetic basis for functional hypogonadotropic hypogonadism. Indeed, this is the first time that this has been demonstrated as a casual mechanism in delayed puberty.

This is a beautiful manuscript with important data to better understand patients with delayed puberty and hypogonadotropic hypogonadism. With the study, very elegant methodology was used. It is well written and easy to read.

Comments:

1. The Introduction should be shortened. Background on IGSF10 should be included in this section.
2. After whole exome sequencing and targeted exome sequencing, the authors found 4 mutations (all of them are heterozygous missense variants predicted to be deleterious by {greater than or equal to}3/5 prediction tools) in IGSF10 (3 of them present in public databases). This is important information; however, with what certitude are these variants pathological?
3. To your knowledge, what kind of differences can be established between mutations in IGSF10 and IGSF1 genes in relation with delayed puberty?
4. Do the authors see any differences in the phenotype between patients with IGSF10 mutations and patients with mutations in KAL1 or PROKR2?
5. In table III the characteristics of Delayed Puberty Probands indicate that the sex is predominantly males (9 out of 10). Any specific comment about the only female subject? Regarding estradiol levels in males, did you measure them?
6. It looks like the induction of puberty was done only in 5 patients. Could the authors comment on the response and the degree of puberty obtained?
7. It would be of interest to include in Table IV data regarding the final height in the patients if you have it.

1st Revision - authors' response

01 March 2016

***** Reviewer's comments *****

Referee #1 (Comments on Novelty/Model System):

The study presents novel findings that will help in the diagnosis of delayed puberty in patients and will contribute to our understanding of the mechanisms governing GnRH migration and development.

Referee #1 (Remarks):

In the present study, Howard and colleagues present a compelling series of human genetic, in vitro and in vivo studies that elegantly describe the novel role of IGSF10 in the migration of GnRH neurons to the hypothalamus during the embryonic period. Their data is supported by a large analysis of patients suffering hypogonadotropic hypogonadism and the function of this molecule postulated and assessed by in vitro models using GN11 cells, which showed reduced migration after Igsf10 knockdown, and reduced migration of GnRH neurons in zebra fish with a morpholino knockdown approach of this molecule. Overall, the study is innovative, informative, well designed and the results clearly stated. There are only a few comments regarding the proposed mechanism of action for Igsf10:

- The authors tested the migration of GN11 cells after KD of *Igsf10*. This experiment assumes that GnRH neurons express *Igsf10*, which would be acting, perhaps, in an auto synaptic feedback loop in GnRH neurons. Still, the authors showed the expression of *Igsf10* in other hypothalamic areas and, in the discussion, mentioned that this molecule probably participates in the creation of a gradient needed to direct GnRH neuronal migration. It is therefore not clear whether GnRH neurons may also express this molecule or whether GN11 cells, due to their immortalized nature, are not a faithful replication of GnRH neurons in vivo. It would be good if the authors clarified this by assessing the expression of *Igsf10* in other GnRH cell lines and, if possible, through double label ISH with better resolution than the images presented in Figure 4.

The in vitro migration assay shown in the study involves shRNA knockdown of Igsf10 in NIH3T3 cells, a mouse fibroblast derived cell line, which we have shown to have high endogenous expression of Igsf10. The evidence from our in situ hybridization studies of Igsf10 expression in the nasal mesenchyme led to the hypothesis that Igsf10 acts as a chemokine to influence GnRH neuronal migration. Thus we did not anticipate that GnRH neurons or GN11 cells would express Igsf10 and did not attempt to knockdown Igsf10 in GN11 cells. Instead, we used NIH3T3 cells as a model of nasal mesenchyme tissue to demonstrate that knockdown of Igsf10 in these cells leads to reduced migration of the GN11 cells plated alongside, in comparison to those plated alongside NIH3T3 cells with normal Igsf10 expression. The manuscript has been modified to clarify this experimental set-up

Lines 267-272: 'We performed co-culture experiments of GN11 aggregates placed on confluent NIH3T3 monolayers. NIH3T3 cells, derived from a mouse embryonic fibroblast cell-line, express high levels of endogenous Igsf10. The NIH3T3 cells were treated with scrambled- or Igsf10-shRNAs, the latter leading to highly reduced levels of Igsf10 expression (Fig 5A).'

- The authors nicely explain that the amount of mutations accumulated in a single individual may account for the different magnitudes in the HH phenotype observed, however, this does not explain the adult onset of HH discussed in lines 321+. If the role of *Igsf10* is solely in the migration of GnRH neurons, as suggested by the disappearance of its expression in late embryonic phases, why would these mutations induce secondary amenorrhea after the HPG axis has been properly activated during puberty? Do they know whether this molecule has further developmental regulation? Would it be possible that its expression increases again at the time of puberty onset and/or is regulated by sex steroids in adulthood? Including a few samples from mice at critical developmental time points (e.g. infantile, juvenile, peripubertal and adult) would address this question.

An overlapping phenotype between DP and HA has been seen in a previous study (Caronia et al, NEJM 2011, DOI: doi: 10.1056/NEJMoa0911064), referred to in our manuscript in lines 408-409. The same authors have proposed the mechanism that reduced GnRH neuronal numbers (caused by mutations in e.g. KAL-1 or PROKR2) may lead to both HH and HA. We hypothesise that mutations in IGSF10 may also lead to reduced numbers and/or late arrival of GnRH neurons at the hypothalamus during embryonic life, resulting in defective functionality of the GnRH neuroendocrine network. This mechanism may therefore lead to either delayed onset of puberty or reduced capacity of the HPG axis to respond in times of compromise e.g. in excessive exercise, reduced caloric intake or other stressors that cause HA. Unfortunately we do not have definitive data on the timing of puberty of our HA patients to discover whether they also had DP, but the study by Caronia et al reports DP in 25% of patients with HA. Interestingly, some partial forms of IHH may lead to normal timing of puberty but arrested puberty or infertility later in life, also suggesting that defects in GnRH neuronal function may present after puberty onset. Additionally, we have previously carried out expression studies using in situ hybridisation in peri-pubertal mice and did not see any expression of Igsf10 in peri-pubertal mouse hypothalamus.

The manuscript discussion has been modified to give further detail in response to this point –

Lines 412-416: 'Specifically, this clinical variability can result from mutations in genes such as KAL1, PROKR2 and now IGSF10, which may lead to late arrival or reduced numbers of GnRH neurons to the hypothalamus, thus compromising the function of the GnRH network.'

- Line 193: do they mean "presence" instead of absence?

Filtering of variants based on the exclusion of those found in public databases is a frequently used step in the filtering pipeline in rare diseases. However, in a more common condition such as delayed puberty this filtering step is not appropriate, as we expect to find 'disease-causing' mutations in the public databases, as we find individuals with delayed puberty in the public databases. Thus it is the 'absence' of these variants that cannot be used as one of the exclusion criteria.

- Line 197: One of the mutations has less than 3 D and would be therefore "possibly damaging" according to their criteria.

Damaging or possibly damaging were both assessed as 'deleterious' by our pipeline, but this clarification has been included in the revised manuscript –

Line 184-6: 'All four IGSF10 variants are heterozygous missense variants predicted to be deleterious, damaging or possibly damaging by $\geq 3/5$ prediction tools (Table 2).'

- Figure 4: The data depicting IGSF10 expression in the human tissue is too weak. Are they sure this is specific? Please, include controls using the sense probe in the supplementals.

Panel N is the expression image for the sense probe for human IGSF10, which shows no visible expression, as compared to the purple-colour nasal mesenchyme staining for IGSF10 seen in panels K, L and M.

- Figure 4: Please, include a scale bar in each panel.

Included in revised manuscript.

Referee #2 (Comments on Novelty/Model System):

The authors demonstrate that IGSF10 regulates embryonic GnRH neuronal migration and mutations result in delayed puberty. This is, indeed, a new concept in Molecular Medicine. The manuscript has a high technical quality and the information is novel.

Referee #2 (Remarks):

Sasha et al, with the corresponding author being Prof. Dunkel, present an elegant multinational study where they have identified mutations in IGSF10 in 6 unrelated families, resulting in intracellular retention of this protein, thus with failure in the secretion of mutant proteins. Furthermore, the authors show that knock out of IGSF10 caused reduced migration of immature GnRH neurons in vitro and perturbed migration and extension of GnRH neurons in a *gnrh3:EGFP* zebrafish model. Furthermore, loss-of-function mutations in IGSF10 were identified in patients with hypothalamic amenorrhea. The authors conclude by saying that mutations in IGSF10 cause delayed puberty in humans with common genetic basis for functional hypogonadotropic hypogonadism. Indeed, this is the first time that this has been demonstrated as a casual mechanism in delayed puberty.

This is a beautiful manuscript with important data to better understand patients with delayed puberty and hypogonadotropic hypogonadism. With the study, very elegant methodology was used. It is well written and easy to read.

Comments:

1. The Introduction should be shortened. Background on IGSF10 should be included in this section.

Please find the introduction shortened as requested. We do believe, however, that discussion of IGSF10 should not appear in the introduction, as this is the main discovery of the study and so details of this gene would logically follow in the results section. IGSF10 was not a candidate gene prior to the start of the study, and the study design was based on an unbiased analysis of WES data, which makes it difficult to highlight one gene in the introduction.

We feel that earlier disclosure would preempt this exciting discovery in the results section, and interrupt the flow of the argument. We are happy to take further guidance from the editor on this point.

2. After whole exome sequencing and targeted exome sequencing, the authors found 4 mutations (all of them are heterozygous missense variants predicted to be deleterious by {greater than or equal to}3/5 prediction tools) in IGSF10 (3 of them present in public databases). This is important information; however, with what certitude are these variants pathological?

Our in vitro data demonstrates failure of secretion of the two N-terminal mutations and retention within the intracellular compartment, which we believe shows clear evidence of their pathogenicity. These mutations were found in 6 out of the 10 families identified. It has not been possible to reproduce these studies for the two C-terminal mutations despite many months of trying to express the full-length and C-terminal protein in mammalian cells. Thus, although our in silico predictions give evidence for predicted pathogenicity of the two C-terminal variants, we are not able to conclusively show these to be pathogenic, as we declare in lines 339-340: 'We have identified an additional two rare variants of unknown significance in 4 further families.'

3. To your knowledge, what kind of differences can be established between mutations in IGSF10 and IGSF1 genes in relation with delayed puberty?

None of our patients with IGSF10 mutations had thyroid abnormalities or other features of the IGSF1 syndrome, apart from delayed onset of puberty. One previous publication describes the sequencing of IGSF1 in families from our DP cohort – Joustra et al, Eur J Pediatr 2015 (DOI 10.1007/s00431-014-2445-9). No pathogenic variants in IGSF1 were found in our cohort with 'simple' delayed puberty, again suggesting that mutations in IGSF1 do not cause self-limited delayed puberty in the absence of other features of the IGSF1 syndrome.

4. Do the authors see any differences in the phenotype between patients with IGSF10 mutations and patients with mutations in KAL1 or PROKR2?

All three of these genes are important in early development and specifically for the correct migration of GnRH neurons to the hypothalamus during embryonic life. Mutations in all three are seen in families segregating with HA, DP, and in the case of KAL1 and PROKR2, IHH or KS. However, we have not conclusively demonstrated as yet that mutations in IGSF10 alone lead to IHH or KS. Please see the addition to the discussion in the main text to further emphasise this point (lines 412-416): 'Specifically, this clinical variability can result from mutations in genes such as KAL1, PROKR2 and now IGSF10, which may lead to late arrival or reduced numbers of GnRH neurons to the hypothalamus, thus compromising the function of the GnRH network.'

5. In table III the characteristics of Delayed Puberty Probands indicate that the sex is predominantly males (9 out of 10). Any specific comment about the only female subject? Regarding estradiol levels in males, did you measure them?

We have previously demonstrated that although there is referral bias in the probands seen in the clinic, exploration of their extended families demonstrates a near equal gender distribution i.e. 1.2males:1female (Reference: Wehkalampi et al, J Clin Endocrinol Metab 2008, DOI 10.1210/jc.2007-1786). Our finding of 9 of 10 of the probands with IGSF10 mutations, but a total of 8 male relatives with DP and 6 female with DP with pathogenic IGSF10 mutations, is consistent with this. We do not believe that IGSF10 mutations are more commonly associated with males and there were no specific phenotypic attributes in the one female proband identified. Estradiol was not routinely measured in male patients.

6. It looks like the induction of puberty was done only in 5 patients. Could the authors comment on the response and the degree of puberty obtained?

In our large DP cohort, approximately 50% of patients chose induction. As part of the protocol for diagnosis of self-limited DP, all patients were followed up off treatment until full development (Tanner G4+) was achieved. The manuscript has been amended to add this clarification (lines 447-

450): *'In the 50% of patients from the cohort who choose to have pubertal induction via the use of exogenous sex steroids, all patients were followed up once off treatment until the point of full pubertal development (Tanner stage G4+ or B4+) to ensure pubertal development did not arrest off treatment.'*

7. It would be of interest to include in Table IV data regarding the final height in the patients if you have it.

This has now been included as an extra column in Table IV, and in the main text lines 207-208: 'At adult height, all but 2 probands (3.III.2 and 6.II.1) fell within normal limits for distance to target height (Table 4) (Saari et al, JAMA Pediatr 2015 DOI: 10.1001/jamapediatrics.2015.25).'

Corresponding Author Name: Professor Leo Dunkel

Manuscript Number: EMM-2016-06250-T